# Agreement between antenatal gestational age by ultrasound and clinical records at birth: A prospective cohort in the Brazilian Amazon

**Bárbara Hatzlhoffer Lourenço**[1]*, **Daniel Leal Lima**[2], **Edwin Vivanco**[2], **Rachelle de Brito Fernandes**[3], **Mirian Duarte**[4], **Paulo Augusto Ribeiro Neves**[1,5], **Marcia Caldas de Castro**[6], **Marly Augusto Cardoso**[1], on behalf of the MINA-Brazil Study Group¶

1 Department of Nutrition, School of Public Health, University of São Paulo, São Paulo, Brazil, 2 Juruá Women's and Children's Hospital, Cruzeiro do Sul, Brazil, 3 Municipal Health Secretariat of Cruzeiro do Sul, Cruzeiro do Sul, Brazil, 4 Private Practice in Obstetrics and Gynaecology, São Paulo, Brazil, 5 Postgraduate Program in Epidemiology, Federal University of Pelotas, Pelotas, Brazil, 6 Department of Global Health and Population, Harvard T. H. Chan School of Public Health, Boston, MA, United States of America

¶ Membership of the MINA-Brazil Study Group is listed in the Acknowledgments.
* barbaralourenco@usp.br

**Data Availability Statement:** Data are available upon request due to restrictions of the Ministry of Health of Brazil for sharing data with information

## Abstract

This study aimed to assess agreement between antenatal estimates of gestational age by ultrasound and clinical records at birth in the Brazilian Amazon. Ultrasound examinations were scheduled during the second trimester for 578 pregnant women prospectively screened at primary health care units, following a standardized protocol for image quality control. A multistage algorithm was used to assess the best estimate of gestational age during the antenatal period, considering reliability of last menstrual period (LMP) and acceptable differences in relation to ultrasound estimates derived from fetal biparietal diameter and femur length. Agreement of antenatal estimates of gestational age confirmed by ultrasound and clinical records at birth was analyzed with Bland-Altman plots and kappa coefficients (preterm and postterm births). Overall, ultrasound examinations presented high quality (>90% of satisfactory images), and were adopted as the best estimate of gestational age among 83.4% of pregnant women, confirming reliable LMP in the remaining proportion. On average, difference in gestational age between antenatal estimates and clinical records was 0.43 week (95% CI: 0.32, 0.53). Classification of preterm births had a good agreement (kappa: 0.82, p<0.001), but a poor performance was observed for postterm births (kappa: -0.06, p = 0.92). Higher differences in gestational age were noted for participants with >11 years of education and cases of caesarean deliveries. In conclusion, high-quality ultrasound images from the second trimester of pregnancy based the assessment of gestational age, while reliability of LMP was limited. Information from clinical records at birth presented an acceptable agreement on average and for classification of preterm births, which is relevant for properly interpreting perinatal outcomes. Discrepancies in caesarean deliveries may warrant further investigation.

on patients attending the health services. For the present analysis, delivery data were collected in the Women's and Children's Hospital of Juruá Valley, Acre State Secretary of Health. Data requests can be sent to Marly Augusto Cardoso, principal investigator of the MINA-Brazil Study Group, at marlyac@usp.br, or to the Human Ethical Review Board of the School of Public Health, University of São Paulo, Brazil (http://sites.usp.br/fsp/en/contato/; e-mail: cpq@fsp.usp.br).

**Funding:** The present study was funded by the Brazilian National Council of Technological and Scientific Development, CNPq (http://cnpq.br/ – grant 407255/2013-3; MAC, MCC), the São Paulo Research Foundation, FAPESP (http://fapesp.br/ – grant 2016/00270-6; MAC) and the Maria Cecilia Souto Vidigal Foundation (http://www.fmcsv.org.br/ – MCC). The views expressed in the present article are those of the authors and not necessarily those of any funding agencies. The funders had no role in study design, data collection and analysis, decision to publish, or preparation of the manuscript.

**Competing interests:** The authors have declared that no competing interests exist.

## Introduction

Gestational age (GA) is essential to interpret and manage maternal and infant health indicators until time of delivery and also after birth [1, 2]. Implications span from the individual level to population and public health purposes, allowing for proper scheduling of antenatal care, evaluation of maternal weight gain during pregnancy, monitoring of fetal growth parameters and timely clinical decision-making in the perinatal period. In addition, such information aids measuring the occurrence and burden of preterm and postterm births, as well as understanding risk factors for perinatal morbidity and mortality, enabling proposition of interventions and public policies. Accurate assessment of GA, nevertheless, imposes some challenges [2].

Ultrasound-based dating techniques in the first trimester of pregnancy (<14 weeks) are currently deemed the most accurate method to establish or confirm GA, through the measurement of fetal crown-rump length up to 84 mm [1, 3]. Estimates derived from self-reporting on the first day of last menstrual period (LMP), in turn, are widely accessible to calculate duration of pregnancies [4]. While important limitations regarding regularity of menstrual cycles, use of hormonal contraceptives and recall bias should be considered [2], recording of LMP in clinical records and use of this data in epidemiological studies seldom indicates its reliability. Reasonable comparability of certain LMP with GA estimated by crown-rump length has been shown when information was collected very early in pregnancy, at around 5 to 8 weeks [5], and also if assisted by home calendars for prospective annotation of dates with periodic surveillance through urine-based testing for pregnancy [6].

Low- and middle-income areas worldwide are characterized by inequities in structure and access to education and health services [7]; despite progress since 1990, overall coverage of early antenatal care visits was only 48.1% in developing regions in 2013 [8]. Based on reference early ultrasounds, recent evidence from Papua New Guinea, Guatemala and Brazil, for example, pointed out that mid-pregnancy ultrasound examinations, with a combination of fetal biometric parameters, are still superior in determining GA in comparison with LMP or neonatal assessment methods such as Ballard or Capurro [9–11]. Concurrently, however, adequate description of standardized procedures and image quality control in ultrasound examinations, especially relevant concerns for research in these settings, are yet scarce in the literature.

There is important indication from large birth register-based studies conducted in developed countries that great discrepancy in estimates of GA might negatively impact on pregnancy, delivery and neonatal outcomes, including associations with higher rates of pre-eclampsia, gestational diabetes, birth asphyxia and fetal and infant mortality [12, 13]. These consequences could be particularly critical in resource-poor areas, which are subject to more intense adverse effects of environmental, socioeconomic and lifestyle aspects on perinatal health.

As part of a prospective cohort conducted in the Brazilian Amazon area, the present study aimed to describe procedures for GA assessment among participants with follow-up since the antenatal period, encompassing image quality control of ultrasound examinations performed in mid-pregnancy and evaluation of reliability of LMP data. We also performed an agreement analysis of the best estimate of GA by ultrasound during the antenatal period in relation to information registered in clinical records at birth.

## Methods

### Study design and population

This is a prospective study in Cruzeiro do Sul, Acre, Brazilian Amazon (latitude: 07º 37' 52" S; longitude: 72º 40' 12" W), part of the "MINA-Brazil–Maternal and Child Health and Nutrition

in Acre, Brazil" Study [14]. The city has approximately 80,000 inhabitants, half of them women, and its Human Development Index was considered medium by 2010, at 0.664, which is below the national average [15].

Our study population was composed of pregnant women enrolled in antenatal care in all primary health care units from the urban area (n = 13). Women up to 20 weeks of pregnancy, as initially screened by their LMP at the primary health care units, who were living in the city and intended to deliver at the only maternity hospital in Cruzeiro do Sul were considered eligible for this study. We estimated to track approximately 850 pregnant women, considering: (i) the number of deliveries at the local maternity hospital in 2013 (n = 1,780), (ii) the percentage of urban population in the municipality (around 60%), and (iii) the coverage for local primary health care services (equivalent to 80%). From February 2015 to January 2016, all eligible women were screened on a weekly basis and invited to participate in our study through phone calls or home visits. Exclusion criteria included incorrect screening according to LMP at the primary health care units, moving to the rural area or other cities and miscarriages before providing informed consent to participate. The Human Ethical Review Board of the School of Public Health of the University of São Paulo approved the research protocol (process 872.613, November 13, 2014). All procedures were in accordance with the 1964 Helsinki declaration and its later amendments. Written informed consent was obtained from each participant or a legal guardian for adolescents.

## Fieldwork procedures and data collection

Trained fieldworkers conducted face-to-face sociodemographic and health interviews with women up to 20 weeks of pregnancy, gathering information on maternal age, skin color and education level, assistance from conditional cash transfer program, living or being married to a partner, being pregnant for the first time, planning for the current pregnancy and attending antenatal care. Additionally, data on regularity and usual duration of menstrual cycles (regular if usually lasting from 24 to 32 days), as well as type of contraceptive methods in use before conception (if in use, and type of method) were collected.

Subsequently, ultrasound examinations were performed in up to two follow-up assessments per participant during the antenatal period. The first assessment was tentatively scheduled between 16 to 20 weeks of pregnancy based on LMP. Considering characteristics of the study setting on coverage of antenatal care [14], second trimester ultrasound examinations were deemed as more feasible to confirm GA in comparison with CRL. Three field physicians with high-quality training in ultrasound examinations (DLL, EV, RBF) were presented to the study equipment and protocols for image acquisition, storage and extraction. In each ultrasound examination, field physicians were aware of participant's LMP and required to ascertain fetal biometric parameters in cephalic (biparietal diameter, occipito-frontal diameter and head circumference), abdominal (transverse abdominal diameter, anterior-posterior abdominal diameter and abdominal circumference) and femoral (femoral length) planes, as well as to document the volume of amniotic fluid, placental localization and fetal presentation. Visualization of landmarks in cephalic, abdominal and femoral planes for correct measurement followed procedures as detailed by Papageorghiou et al. [16], with placement of callipers and ellipses considering the outer bone edges ("outer to outer" measurements). Also, each plane was assessed according to quality criteria for image acquisition using a self-scoring system [17], as shown in Table 1. In the present study, images of cephalic and abdominal planes were considered satisfactorily obtained when scored ≥5 points out of a maximum of 6 points, and images of femoral plane were deemed properly acquired when scored ≥3 points out of a maximum of 4 points. All biometric variables of each plane were ascertained in a same

**Table 1. Scoring criteria for ultrasound image acquisition during follow-up assessments throughout the antenatal period in the MINA-Brazil cohort study.**

| Plane | | |
|---|---|---|
| **Cephalic** | **Abdominal** | **Femoral** |
| Symmetrical plane | Symmetrical plane | Both ends of the bone clearly visible |
| Thalami visible | Stomach bubble visible | Femur occupying ≥30% of total image size |
| Cavum septi pellucidi visible | Portal sinus visible | Angle <45º to the horizontal |
| Cerebellum not visible | Kidneys not visible | Callipers correctly placed |
| Head occupying ≥30% of total image size | Abdomen occupying ≥30% of total image size | |
| Callipers and ellipse correctly placed | Callipers and ellipse correctly placed | |
| **Score for classification of satisfactory images** | | |
| ≥5 points out of 6 points | ≥5 points out of 6 points | ≥3 points out of 4 points |

2-dimensional image. A portable SonoSite TITAN machine (SonoSite Inc., Bothell, WA, USA) was used in all ultrasound examinations and values for all biometric parameters were recorded with the support of a research assistant. Generated by the SonoSite package software, the corresponding GA for measurements of fetal biparietal diameter and femur length [18] from the first available follow-up assessment were used to estimate a mean ultrasound GA.

For external quality control, ultrasound examinations were submitted on a monthly basis to an independent expert obstetrician based in São Paulo (MD) for blinded scoring according to the same quality criteria for image acquisition (Table 1). In the first semester of follow-up assessments during the antenatal period, all exams were re-evaluated; in subsequent months of assessments, samples with 30% of exams conducted per field physician every four weeks were randomly selected for quality control. Of these exams, three image files of the cephalic plane and one image file of the femoral plane were not properly stored and could not be scored by the expert obstetrician. Monthly individual reports on quality of image acquisition were shared with each field physician to maintain proper performance, which was set at <10% of low scoring images.

At the maternity hospital, research assistants regularly assessed clinical records to collect information on outcome of pregnancy (live birth, miscarriage, stillbirth), GA and type of delivery (vaginal or caesarean). All data were collected using personal digital assistants and tablets programmed with the Census and Survey Processing System–CSPro versions 4.1 and 6.1 (U.S. Census Bureau, Washington, DC, USA).

## Data analysis

Basic sociodemographic and health data related to the antenatal period were described for study participants at baseline. Afterwards, confirmation of GA considered the mean ultrasound estimate of GA in relation to information on LMP, regularity of menstrual cycles and type of contraceptive methods in use before conception, which included condoms, oral contraceptive pills, contraceptive injections, hormonal intrauterine devices and emergency contraceptive pills, in isolate or combined use. Contraceptive methods were categorized as non-hormonal or hormonal, as the latter may affect the reliability of self-reported LMP. Multiple pregnancies and stillbirths were additionally excluded from this analysis. No participant referred use of assisted reproductive technology for conception.

Initial analysis of the ultrasound examinations followed a two-step procedure. First, self-scoring of quality criteria for satisfactory image acquisition of cephalic and femoral planes were observed for each field physician. Second, from the scores generated in the re-evaluation

conducted by the independent expert obstetrician, proportions of satisfactorily acquired images of cephalic and femoral planes were compared among field physicians with chi-square or Fisher's exact tests.

Then, a multistage algorithm was used to assess the best estimate of GA, according to an evaluation of reliability of LMP and acceptable differences in relation to ultrasound examinations. LMP was considered reliable if reported with certainty, along with regular menstrual cycles and no hormonal contraceptive use before conception [19]. After a review of guidelines and related literature [1, 11, 20–24], acceptable differences of reliable LMP in relation to the ultrasound estimate were defined at ≤7 days for ultrasound examinations conducted up to 22 weeks of pregnancy; for pregnancies with longer durations, differences were acceptable at ≤14 days. Regarding timing of ultrasound data acquisition in the present study, 62.9% of participants were examined up to 20 weeks of pregnancy and 81.9% up to 22 weeks of pregnancy; only 5.0% of examinations were conducted in the third trimester. Such criteria were additionally compared with other definitions for acceptable differences between LMP and ultrasound estimates [1, 21], and our findings were not substantially changed. Among 14 participants, a fetal crown-rump length <84 mm was noted at the moment of the first ultrasound examination during follow-up assessments and the corresponding GA was set as the best available estimate in the antenatal period. One additional participant did not provide information on menstrual cycles and contraceptives; therefore, the ultrasound estimate of GA was solely used.

Finally, agreement of GA in weeks according to antenatal estimates by ultrasound and clinical records at birth was analyzed. For all participants with complete data and live births from singleton pregnancies, these sources of information were compared with calculation of kappa coefficient considering classification of preterm (<37 weeks of pregnancy) and postterm (≥42 weeks of pregnancy) births. Agreement was defined as good if kappa >0.80 and as substantial if kappa >0.60 [25]. In relation to final GA in weeks as a continuous variable, Bland-Altman plots were used. The Y-axis displays the difference between antenatal estimates by ultrasound and clinical records at birth and the X-axis represents the average of these two measurements [26]. Mean difference and its 95% confidence interval (95% CI) were provided along with limits of agreement. Linear regression of differences on averages was fitted to investigate proportional bias, detected when the coefficient is significantly different from zero. Bland-Altman analyses were also stratified by categories of maternal, antenatal and birth characteristics of interest. All analyses were performed using Stata 11.2 (Stata Corp., College Station, TX, USA).

## Results

### General characteristics of study participants

In the present study, 860 pregnant women were screened, of whom 699 were eligible. Among them, 70 participants were not found due to missing information in home address or wrong telephone numbers and an additional 38 subjects declined participation at the initial interview. Out of 588 participants with baseline data (84.1% of eligible), there were four cases of confirmed multiple pregnancies, four miscarriages before the first ultrasound examination and two refusals to attend the ultrasound examinations. The remaining 578 participants with singleton pregnancies were suitable for entering a flow for confirmation of GA during follow-up assessments in the antenatal period (Fig 1).

General characteristics of these participants are shown in Table 2. Mean age was 24.4 years (SD: 6.38), with a third of pregnancies among adolescents and 85.8% were non-white. Most participants had ≥9 years of education and lived or were married to a partner. Almost all participants were attending antenatal care services, notwithstanding only 44.0% of pregnancies were planned.

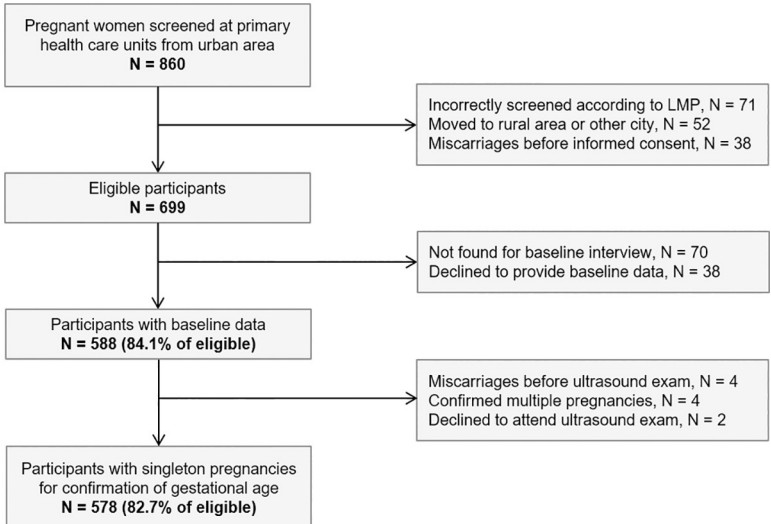

**Fig 1. Flow diagram for eligibility of participants with singleton pregnancies in the MINA-Brazil cohort study.**
LMP: last menstrual period.

## Quality of image acquisition in ultrasound examinations

In up to two follow-up assessments conducted per participant during the antenatal period, three trained field physicians performed 878 ultrasound examinations of singleton pregnancies during second and third trimesters. At the positioning of each biometric plane, field physicians self-scored over 99% of images of both cephalic (n = 873) and femoral (n = 875) planes as satisfactory according to the quality criteria for image acquisition (Table 1).

A total of 520 ultrasound exams were submitted to an independent expert obstetrician for blinded external quality control on a monthly basis. The expert obstetrician evaluated 94.0% of the selected cephalic plane images (n = 489) as satisfactorily acquired, with a mean score of

**Table 2. General characteristics of participants with singleton pregnancies in the MINA-Brazil cohort study.**

| Characteristics | N | % |
|---|---|---|
| Age | | |
| <20 years | 173 | 30.0 |
| 20–35 years | 362 | 62.7 |
| >35 years | 42 | 7.3 |
| Skin color | | |
| White | 82 | 14.2 |
| Non-white | 495 | 85.8 |
| Education | | |
| <9 years | 134 | 23.2 |
| 9–11 years | 308 | 53.4 |
| >11 years | 135 | 23.4 |
| Conditional cash transfer program assistance | 225 | 39.0 |
| Living or married to a partner | 445 | 77.1 |
| Pregnant for the first time | 256 | 44.4 |
| Planning for current pregnancy | 254 | 44.0 |
| Attending antenatal care | 553 | 95.8 |

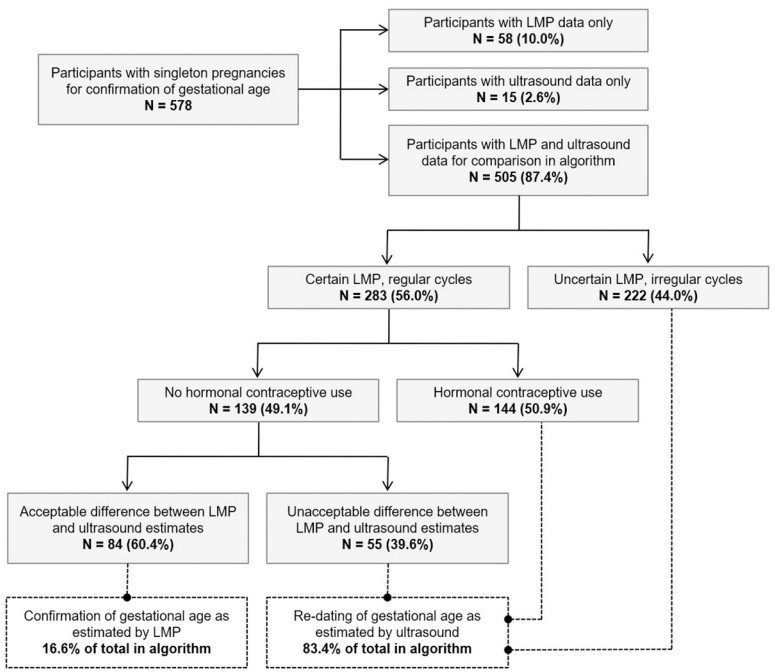

**Fig 2. Multistage algorithm for confirmation of the best estimate of gestational age during the antenatal period among participants with singleton pregnancies in the MINA-Brazil cohort study.** LMP: last menstrual period.

5.63 points out of a maximum of 6 points (median score: 6 points, IQR: 5–6 points). For the femoral plane, 98.9% of images (n = 514) were properly captured, with a mean score of 3.83 points out of a maximum of 4 points (median score: 4 points, IQR: 3–4 points). Performance for adequate acquisition of cephalic or femoral plane images was not significantly different among the three field physicians (p = 0.57 and p = 0.52, respectively).

## Gestational age assessment during the antenatal period

Among 578 participants with singleton pregnancies and follow-up during the antenatal period, 10.0% had information available on LMP only (n = 58) and 2.6% had data exclusively from ultrasound examinations (n = 15) in order to assess the best estimate of GA. For the remaining 505 participants (87.4%), LMP and ultrasound data from the first available follow-up assessment were compared in a multistage algorithm for confirmation of GA, considering regularity of menstrual cycles, type of contraceptive methods in use before conception and difference between LMP and ultrasound estimates, as shown in Fig 2. Following the algorithm, LMP was considered unreliable and therefore re-dated by the ultrasound information for 421 participants. Among an additional 84 participants, LMP was deemed reliable and properly confirmed by data from ultrasound examinations.

## Agreement between antenatal estimates by ultrasound and clinical records at birth

At the maternity hospital, 546 out of 578 participants (94.5%) with singleton pregnancies and antenatal information of GA could be reached. Of these, there were six stillbirths and one miscarriage as outcomes. Antenatal information on GA exclusively relied on LMP, without confirmation by ultrasound examinations, for an additional 44 participants. Information for GA was missing in clinical records at birth for four participants. Thus, considering all live births from

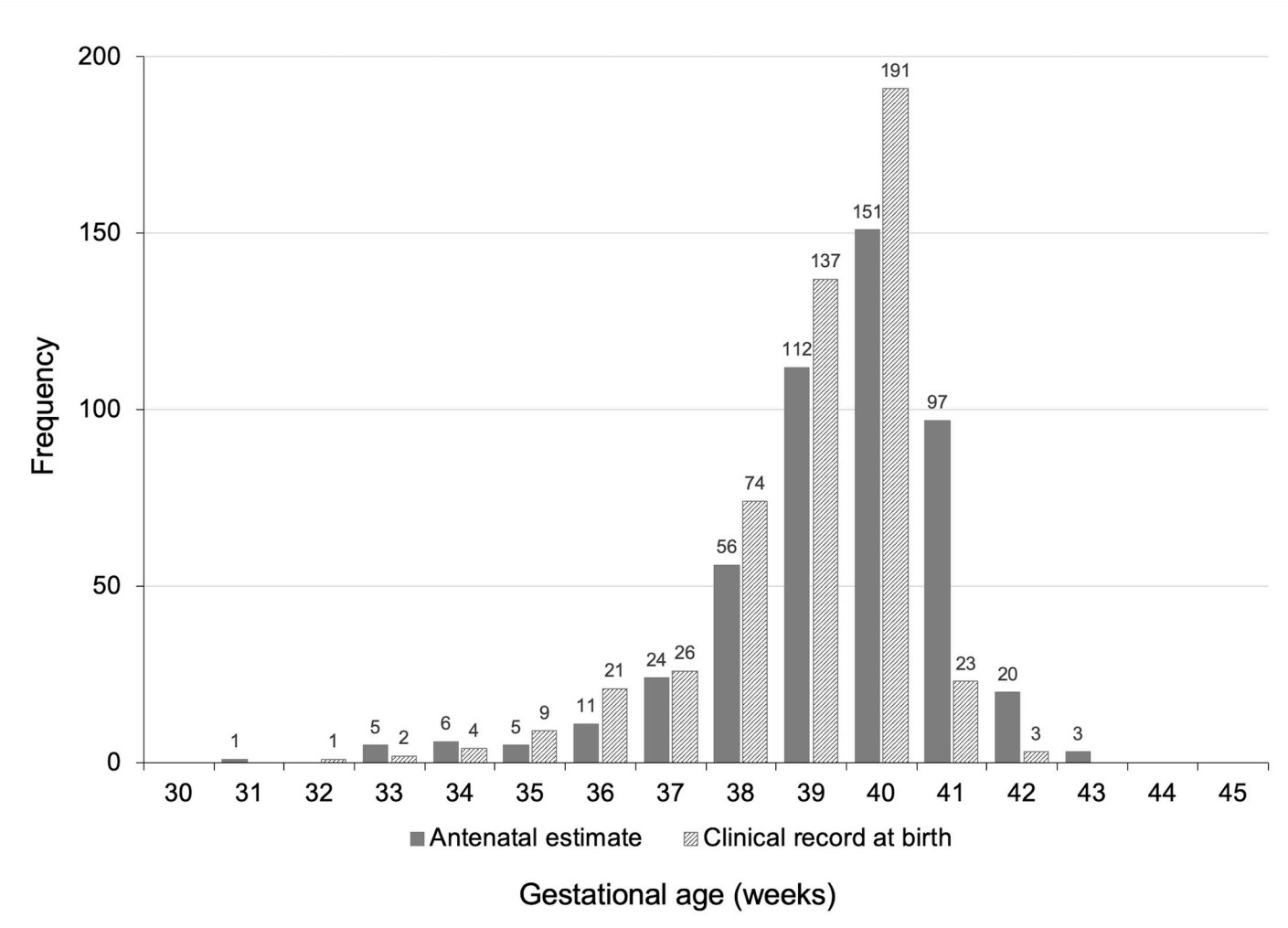

**Fig 3. Distribution of gestational age in weeks according to antenatal estimates by ultrasound and clinical records at birth in the MINA-Brazil cohort study.**

singleton pregnancies, agreement analysis for GA between antenatal estimates by ultrasound and clinical records at birth was conducted for 491 participants (54.0% vaginal deliveries, 48.1% female infants).

Distributions of GA in weeks are presented in Fig 3, with median values of 40 weeks (IQR: 39–40; range: 31–43 weeks) with antenatal estimates by ultrasound and 39 weeks (IQR: 38–40; range: 32–42 weeks) with clinical records at birth. In each histogram, more than 70% of values fell between 39 to 41 weeks. According to antenatal estimates of GA and clinical records at birth, respectively, 5.7% (n = 28) and 7.5% (n = 37) of deliveries were preterm. Classification of preterm births had a good agreement (overall proportion of agreement: 97.8%; kappa coefficient: 0.82, p<0.001). However, classification of postterm births considering antenatal estimates (4.7%; n = 23) and clinical records at birth (0.6%; n = 3) presented poor strength of agreement (overall proportion of agreement: 87.8%; kappa coefficient: -0.06, p = 0.92).

Absolute differences in GA as a continuous variable were within one week for 84.1% of our participants, and within two weeks for 96.1%. Mean GA estimates were significantly higher as defined by antenatal data when compared with clinical records at birth. As indicated by Bland-Altman analysis, difference was on average 0.43 week (95% CI: 0.32, 0.53) and limits of

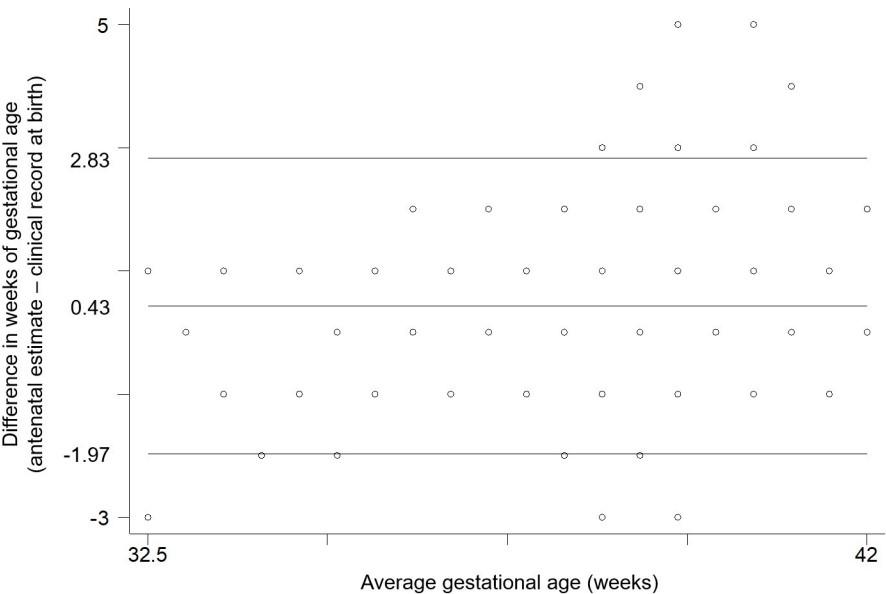

**Fig 4. Bland-Altman plot comparing agreement for gestational age in weeks between antenatal estimates by ultrasound and clinical records at birth in the MINA-Brazil cohort study.**

agreement varied from -1.97 to 2.83 weeks (Fig 4). The linear regression coefficient of difference on mean was equal to 0.18 (95% CI: 0.11, 0.25), indicating significant proportional bias. Visual inspection of the Bland-Altman plot suggests that wider differences were observed at higher mean values of GA between measurements. This finding indicates that, at the higher end of the distribution of GA, clinical records at birth registered lower values compared with antenatal data, which is consistent to some extend with the small proportion of postterm births according to clinical records and the poor agreement between both sources of information for such classification.

Of note, mean difference was lower and not significant (0.13 week; 95% CI: -0.11, 0.36) between the two sources of information in the category of participants with <9 years of education (n = 104). On the other hand, mean difference among participants with >11 years of education (n = 116) was equal to 0.60 week (95% CI: 0.40, 0.79). Similarly, among cases of vaginal delivery (n = 265), a mean difference of 0.22 week (95% CI: 0.07, 0.37; Fig 5A) was significantly smaller than that observed for caesarean sections (n = 226), equivalent to 0.67 week (95% CI: 0.53, 0.81; Fig 5B) when comparing antenatal data with clinical records at birth. It is interesting to notice that maternal education was significantly associated with type of delivery in our study. Caesarean deliveries occurred among 35.6% of women with <9 years of education, 45.6% of women with 9–11 years of education, and reached 56.9% of those with >11 years of education (p = 0.006).

Significant variations in the magnitude of agreement between the two sources of information for GA were not detected according to categories of maternal age and skin color, assistance from conditional cash transfer program, living or being married to a partner, being pregnant for the first time, planning for the current pregnancy, or infant's sex.

## Discussion

In this prospective study conducted in the Brazilian Amazon area, we relied on ultrasound examinations in the second trimester of pregnancy to confirm the best estimate of GA during

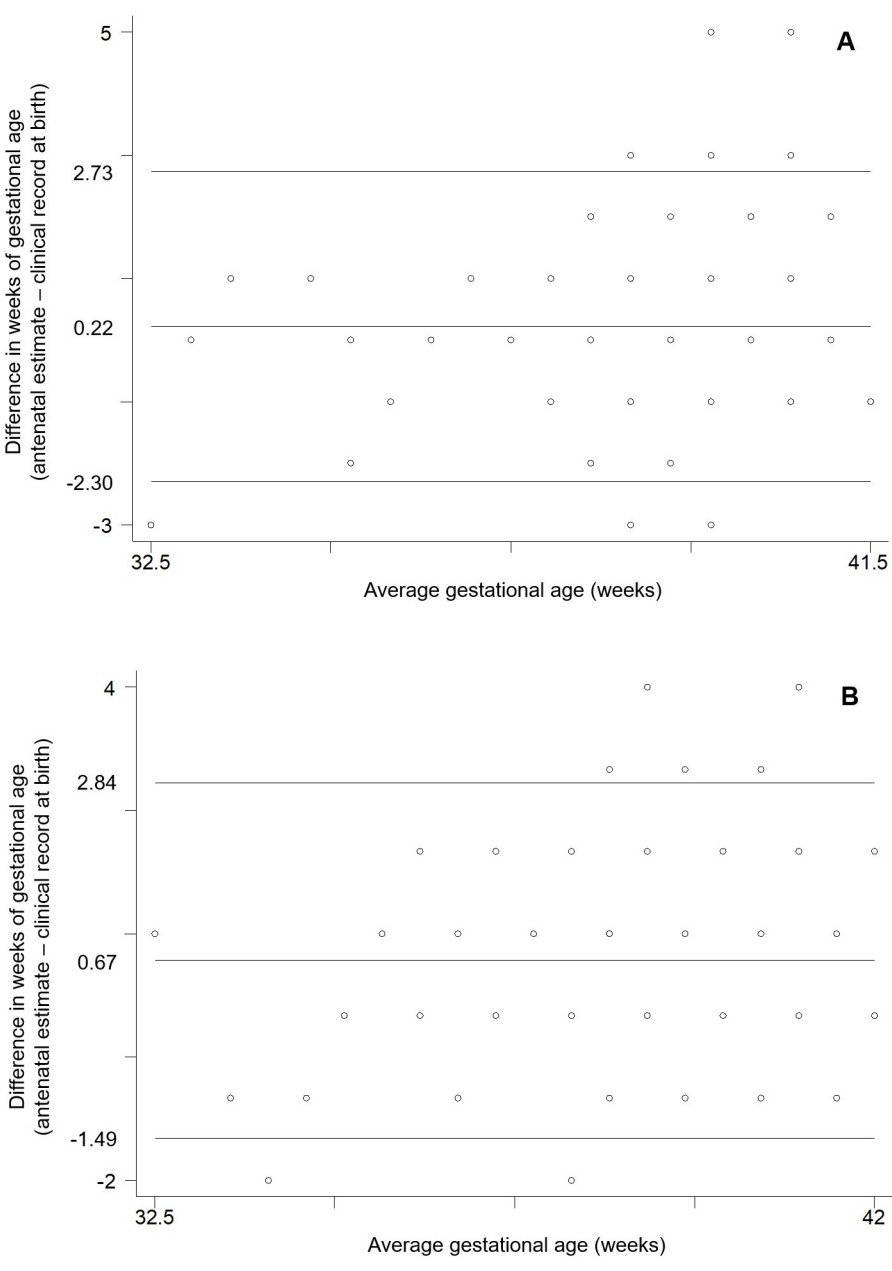

**Fig 5.** Bland-Altman plots comparing agreement for gestational age in weeks between antenatal estimates by ultrasound and clinical records at birth according to type of delivery, vaginal (A) and caesarean section (B), in the MINA-Brazil cohort study.

the antenatal period, combining measures of fetal biparietal diameter and femur length in relation to reliable LMP information. Images of both cephalic and femoral planes presented consistently high quality, among all field physicians and also in external evaluation by an independent expert obstetrician, with >90% of satisfactorily acquired images. Agreement analysis indicated that antenatal estimates of GA by ultrasound differed on average by 0.43 week (95% CI: 0.32, 0.53), or approximately 3 days, when compared with clinical records at birth. Proportional bias was found, suggesting that wider differences between these sources of information for GA were observed at higher mean values of GA between measurements.

Our procedures for GA assessment considered important characteristics of our study setting, which are shared with many low- to middle-income areas. Screening of participants in the urban area of Cruzeiro do Sul faced a panorama of partial registration of households and provision of primary health care services. Along with other determinants of access and use of antenatal care services, including maternal education and employment, family support and reproductive health status [27], a systematic analysis concluded that antenatal care is still not initiated at the first trimester among more than half of pregnant women from developing countries [8]. In a nationwide sample of postpartum women in Brazil, 26% had ultrasound examinations registered <14 weeks of pregnancy; the lowest proportion (12.3%) was observed among participants from the Northern region of the country [28], where our study setting is located. Therefore, ultrasound examinations in the second trimester should be considered as a significant approach in such regions for estimating GA with relative accuracy [1, 21, 22, 24].

Besides appropriate timing for the use of ultrasound examinations in GA assessment, use of fetal biometric parameters as a proxy of duration of pregnancies is only as reasonable as the quality of images. Establishing proper methodology for their acquisition is consequently crucial, but this practice is not often clearly implemented and reported in many epidemiological studies. For the present analysis, three field physicians with a high previous level of training for ultrasound examinations followed a unified protocol to obtain 2-dimensional images of fetal biometric planes [16]. Standardized visualization of anatomical landmarks and placement of callipers and ellipses in a reproducible fashion minimize possible systematic errors and warrant comparability of our measurements with the literature on determination of GA and fetal growth patterns. Moreover, self-scoring quality criteria [17] were defined a priori for each fetal biometric plane to support satisfactory image acquisition during every ultrasound examination, as conducted in a large multicenter research project for construction of fetal growth standards [29]. Such practice also enabled an independent expert obstetrician to periodically accomplish blinded objective evaluation of performance per field physician on a substantial proportion (around 60%) of our images, consolidating a set of high-quality images for estimation of GA in this study. A randomized controlled trial with 258 ultrasonographers indicated that an audit process based on objective scoring criteria significantly improved image quality over time, particularly among participants who received feedback from an experienced ultrasound professional [30]. Similarly to our findings, standardization and quality control of fetal biometry in ultrasound examinations have been proven feasible in the Intergrowth-21st Project, in which 10% of images were randomly selected for re-evaluation [31].

The debate on methods to establish or confirm GA should include an adequate appraisal of reliability of self-reported LMP in light of some reproductive characteristics before conception [19]. Following a multistage algorithm for confirmation of GA, reliability of LMP in this study was noticeably compromised by uncertainty in self-report, irregular menstrual cycles and use of hormonal contraceptives before conception. As a result, a subsequent comparison with ultrasound data was deemed possible for 28% of all 505 participants in order to check for acceptable differences between both estimates of GA. A prospective ultrasound demonstration project with 177 pregnant women in Malawi confirmed self-reported LMP data with ultrasound examination for almost two thirds of participants [21]. However, and as observed in many other epidemiological studies, there was no indication of compilation of important periconceptional reproductive variables to firstly regard LMP as a dependable source of information for estimating GA. In line with our results, caution was advised for the use of LMP in a hospital-based nationwide study in Brazil assessing validity of antenatal GA estimates [11]. Lee at al. [32] systematically reviewed the literature on the diagnostic accuracy and reliability of GA determination of newborns in relation to ultrasound or LMP. The authors indicated poor quality of studies on GA ascertainment and listed priority actions for low- to middle-income

regions, including improvement of coverage and development of novel ultrasound approaches [32].

In view of a careful process for obtaining timely and satisfactory ultrasound images along with verification of reliability of LMP, and a combination of these data to confirm the best estimate of GA among our participants with follow-up since the antenatal period, clinical records at birth gathered information on GA in fairly good agreement on average. Mean difference between these two sources of information was significantly different from zero, equivalent to 0.43 week (or 3 days) and a vast majority of participants (84.1%) with absolute differences falling within one week. Along with a good agreement for the classification of preterm births (kappa coefficient: 0.82), we may assume this bias has probably overall limited clinical meaning or implications in our study population. The present findings had similar magnitude with the mean difference estimated in 2.8 days between gold-standard crown-rump length in the first trimester (reference method) and prospectively collected LMP (with aid of a home calendar and confirmation of pregnancy with urine testing) in a validation study in rural Bangladesh [6]. In our analysis, clinical records at birth showed superior agreement levels for continuous GA estimates and for classification of prematurity than the mean differences, limits of agreement and kappa coefficients observed for self-reported LMP, information on the onset of fetal movements during pregnancy (quickening), symphysis pubis fundal-height and Ballard newborn assessment among several previous studies conducted in developing countries with reference early ultrasounds for comparison [9, 10, 21].

A couple of aspects in our data deserve further comment. First, though complete and available for virtually all patients, GA from clinical records at birth as collected from the maternity hospital may combine mixed sources of information, and also at different time points in pregnancy among participants. In Brazil, LMP is still the first fill-in option for calculation of GA according to national guidelines on antenatal care [20], but ultrasound methods are increasingly available and advised [11]. Hence, with the present agreement analysis, we were able to compare our antenatal estimates by ultrasound with the measure of GA supplied to health information systems on live births.

Second, as noted in histograms of GA, distribution of values according to clinical records at birth do not appear sufficiently dispersed to the right, remarkably after 40 weeks, in a different pattern in relation to antenatal estimates by ultrasound, which resulted in significant proportional bias and inconsistent agreement for the classification of postterm births in our study. Occurrence of postterm births is reported worldwide at around 5 to 10% of all pregnancies [33, 34], and this figure was more closely approximated by the antenatal estimates of GA calculated for our study population. It could be interesting to discuss such finding within the complex scenario of a prominent high proportion of early term births (37 to 38 weeks) in Brazil, as verified according to GA directly collected from clinical records [35]. As reported by Leal et al., there were 35% of early term births among all live births from 37 to 40 weeks; the care provider initiated 44% of those births, with significant association with caesarean section as the type of delivery [35].

In fact, our analysis depicted significantly higher mean differences in GA according to categories of maternal education and type of delivery–higher levels of education were associated with higher occurrence of caesarean sections among our study participants. As documented in the literature, there are substantial socioeconomic inequalities in caesarean deliveries within low- to middle-income countries, encompassing a spectrum from inadequate access to elective use of this health technology [36–38]. Highly educated women compared to those who were less educated, as well as cases of caesarean sections compared to participants with vaginal delivery, had a somewhat poorer agreement between estimates of GA in our study. Additional caution in using GA information may be required if analyzing data specifically for these

subgroups. Further investigations in this sense remain needed, considering high caesarean sections rates, medical indications involved and the resulting burden for adverse maternal and infant outcomes.

There are some limitations to this study. Due to fieldwork constraints, we were not able to perform a quantitative evaluation of ultrasound images by performing examinations in duplicate or triplicate and calculating intra- and inter-observer variations in biometric measurements. However, using objective scoring criteria, we assumed high cut-off points for each fetal biometric plane to regard an image as satisfactorily acquired. Follow-up during the antenatal period was possible only in primary health care units from the urban area. Although the majority of local population refer to these units, it is difficult to presume the panorama among rural residents. Major strengths include comprehensive procedures for assessing GA during the antenatal period based on ultrasound examinations combined with evaluation of reliability of LMP. We could demonstrate their feasibility for the Brazilian Amazon context through the use of a portable, battery-operated machine, besides establishing a partnership with the local health services and qualifying field physicians in our research protocol. Agreement analysis was based on a longitudinal design, with good follow-up rate and completeness of prospectively collected data.

## Conclusion

In conclusion, among participants of this cohort study in the Brazilian Amazon area, the best estimate of GA during the antenatal period was confirmed with a multistage algorithm combining high-quality ultrasound images in the second trimester of pregnancy and reliable LMP information. GA as collected from clinical records at birth presented an acceptable agreement in relation to the antenatal estimates, on average, and also for the classification of preterm births. These findings are relevant to support the appropriate interpretation of perinatal variables, with a view to the integration of maternal and infant care.

## Supporting information

**S1 File. Multistage algorithm for assessing the best estimate of gestational age during the antenatal period, considering reliability of last menstrual period and acceptable differences in relation to ultrasound estimates.**
(PDF)

## Acknowledgments

The authors are thankful to all participants, professional health workers, and research team members of the MINA-Brazil Study Group involved in fieldwork for this study, as well as the Municipal Health Secretariat, all primary health care units, and the Juruá Women's and Children's Hospital of Cruzeiro do Sul.

Members of MINA-Brazil Study Group are: Alicia Matijasevich Manitto, Barbara Hatzlhoffer Lourenço, Maíra Barreto Malta, Marly Augusto Cardoso [Principal Investigator, marlyac@usp.br], Paulo Augusto Ribeiro Neves (University of São Paulo, São Paulo, Brazil); Suely Godoy Agostinho Gimeno (Federal University of São Paulo, São Paulo, Brazil); Bruno Pereira da Silva, Rodrigo Medeiros de Souza (Federal University of Acre, Cruzeiro do Sul, Brazil); Marcia Caldas de Castro (Harvard T.H. Chan School of Public Health, Boston, USA).

## Author Contributions

**Conceptualization:** Bárbara Hatzlhoffer Lourenço, Marly Augusto Cardoso.

**Data curation:** Bárbara Hatzlhoffer Lourenço.

**Formal analysis:** Bárbara Hatzlhoffer Lourenço.

**Funding acquisition:** Marcia Caldas de Castro, Marly Augusto Cardoso.

**Investigation:** Bárbara Hatzlhoffer Lourenço, Daniel Leal Lima, Edwin Vivanco, Rachelle de Brito Fernandes, Mirian Duarte, Paulo Augusto Ribeiro Neves.

**Methodology:** Bárbara Hatzlhoffer Lourenço, Marcia Caldas de Castro, Marly Augusto Cardoso.

**Project administration:** Marly Augusto Cardoso.

**Writing – original draft:** Bárbara Hatzlhoffer Lourenço.

**Writing – review & editing:** Bárbara Hatzlhoffer Lourenço, Daniel Leal Lima, Edwin Vivanco, Rachelle de Brito Fernandes, Mirian Duarte, Paulo Augusto Ribeiro Neves, Marcia Caldas de Castro, Marly Augusto Cardoso.

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
