## [Decision Letter · Decision Letter 0]

3 Jun 2020

PONE-D-20-08619

Agreement between antenatal gestational age by ultrasound and clinical records at birth: a prospective cohort in the Brazilian Amazon

PLOS ONE

Dear Dr. Lourenco,

Thank you for submitting your manuscript to PLOS ONE. After careful consideration, we feel that it has merit but does not fully meet PLOS ONE’s publication criteria as it currently stands. Therefore, we invite you to submit a revised version of the manuscript that addresses the points raised during the review process.

We look forward to receiving your revised manuscript.

Kind regards,

Professor Kwasi Torpey, MD PhD MPH

Academic Editor

PLOS ONE

Journal Requirements:

2. To meet our criteria of reproducibility, please consider providing full details of the algorithms designed

4. One of the noted authors is a group or consortium [MINA-Brazil Study Group]. In addition to naming the author group, please list the individual authors and affiliations within this group in the acknowledgments section of your manuscript. Please also indicate clearly a lead author for this group along with a contact email address.

Additional Editor Comments (if provided):

Reviewers' comments:

Reviewer's Responses to Questions

**Comments to the Author**

1. Is the manuscript technically sound, and do the data support the conclusions?

Reviewer #1: Yes

Reviewer #2: Yes

2. Has the statistical analysis been performed appropriately and rigorously? 

Reviewer #1: Yes

Reviewer #2: No

3. Have the authors made all data underlying the findings in their manuscript fully available?

Reviewer #1: Yes

Reviewer #2: Yes

4. Is the manuscript presented in an intelligible fashion and written in standard English?

Reviewer #1: Yes

Reviewer #2: Yes

5. Review Comments to the Author

Reviewer #1: Dear editor:

Thank you for inviting me to evaluate the article titled “Agreement between antenatal gestational age by ultrasound and clinical records at birth: a prospective cohort in the Brazilian Amazon”. In this paper, the authors performed a prospective study in Brazilian Amazon area which is a developing district to describe a standardized procedure for obtaining gestational age (GA) with ultrasound examination in mid-pregnancy and LMP data.

The text is well arranged and the logic is quite clear. The methods of statistics, ultrasound image evaluation and follow up are rigorous. However there are some defects needs to be improved so I suggest major revision.

1. As a prospective study, the inclusion criteria has been clarified in the manuscript but the exclusion criteria seems not been listed in manuscript while it may be known from the Figure 1.

2. It is noted that the present study was aimed at low- and middle-income area, then I suggest that the income level of people included be considered as one of the baseline characteristics.

3. Further, it is necessary to clarify the way of conception in manuscript (conceive naturally or by ART).

4. As you mentioned, the crown-rump length (CRL) in the first trimester is a gold-standard for evaluating GA, then why didn’t you measure CRL as a reference method?

5. The regularity of menstruation should be defined.

6. In “data analysis” (page 8, line 139), the types of contraceptive methods in use before conception has been documented but it seems that you only separate the contraception as “hormonal contraceptive use” and “non-hormonal contraceptive use”. I think the emergency contraceptives and short-acting contraceptives are different and the detailed information concerning on contraceptive method should be clarified.

7. A multi-stage algorithm was applied to assess the best estimate of GA during the antenatal period. How did you build this algorithm? Have you done a pilot study?

8. It is necessary to explain how the sample size is calculated.

9. The present study analyzed the agreement of antenatal estimates of GA confirmed by ultrasound and clinical records at birth. But what is basis of GA from clinical records? Is it evaluated from LMP or checked by the CRL in the first trimester?

Reviewer #2: Overall I believe this is a strong manuscript with multiple strengths -- the use of the algorithm to determine the the validity of LMP measures and the data on image quality. However, there are some relatively minor issues that should be addressed before publication:

1) Poor comma usage throughout the manuscript which made it distracting and difficult to understand at times

2) Several sentences needed to be reworded for clarity [L95-97] and [L339-341]

3) The description of the algorithm is a bit confusing/misleading. I really enjoyed the analysis but I think the way it is described doesn't make it clear that the algorithm is validating the utility/accuracy of LMP measures

4) There were 15 participants that only had ultrasound and no LMP. However, it is unclear how they were eligible as eligibility was based on LMP determined gestational age

5) The relationship between education and cesarean section was not unexpected. However, the greater difference in LMP and ultrasound in those groups was. I think the potential reasons for this should be discussed.

6) I think Figure 3 would be easier to interpret if the two histograms were overlaid

7) Was there any proportional bias in the Bland-Altman plots ie a relationship between difference in measures for different gestational ages?

8) I think it would be useful to know how the agreement between LMP and ultrasound estimates differed based on the gestational age at ultrasound

6. PLOS authors have the option to publish the peer review history of their article (what does this mean?). If published, this will include your full peer review and any attached files.

Reviewer #1: No

Reviewer #2: No

---

## [Author Response · Author response to Decision Letter 0]

20 Jun 2020

Comments from Reviewers and Point-by-Point Response

PONE-D-20-08619: Agreement between antenatal gestational age by ultrasound and clinical records at birth: a prospective cohort in the Brazilian Amazon

We would like to thank the Academic Editor and the Reviewers for the careful review regarding our manuscript submitted to PLOS ONE. Certainly, the observations and suggestions have proved very helpful toward improving the present study. In addressing the points raised by the reviewers, the contents of the manuscript have been changed as specified below in a point-by-point response to each comment. Pages and lines indicated in each answer in order to locate the changes in the text refer to the file with the unmarked revised version of the manuscript.

Please do not hesitate to contact us if you have any questions regarding the new version of our manuscript. Thank you very much.

Reviewer #1

1. In this paper, the authors performed a prospective study in Brazilian Amazon area which is a developing district to describe a standardized procedure for obtaining gestational age (GA) with ultrasound examination in mid-pregnancy and LMP data. The text is well arranged and the logic is quite clear. The methods of statistics, ultrasound image evaluation and follow up are rigorous.

We are grateful for the reviewer’s feedback on our manuscript. 

2. As a prospective study, the inclusion criteria has been clarified in the manuscript but the exclusion criteria seems not been listed in manuscript while it may be known from the Figure 1.

Thank you for this observation. As requested, we have included information on exclusion criteria for the study on p. 5, l. 81-83, as follows: “Exclusion criteria included incorrect screening according to LMP at the primary health care units, moving to the rural area or other cities and miscarriages before providing informed consent to participate”.

3. It is noted that the present study was aimed at low- and middle-income area, then I suggest that the income level of people included be considered as one of the baseline characteristics.

In the present study, we have gathered information about the ownership of selected assets (such as television, microwave, washing machine, car, among others) to generate a household wealth index through principal component analysis, as broadly used by the Demographic and Health Surveys Program (as discussed at: https://dhsprogram.com/topics/wealth-index/). The household wealth index is usually divided into quintiles to allow the comparison of variables of interest in relation to this survey-specific measure of cumulative living standards among participants. Isolate, the wealth index is not quite informative in describing the baseline characteristics of participants. As alternative measures of socioeconomic status that may be readily comparable with other studies, in Table 2 we have included information on educational level of pregnant women and assistance from a conditional cash transfer program. On p. 5, l. 70-71, we have also provided information on the Human Development Index of Cruzeiro do Sul, as follows: “Its Human Development Index was considered medium by 2010, at 0.664, which is below the national average”. 

4. Further, it is necessary to clarify the way of conception in manuscript (conceive naturally or by ART).

To clarify this point in the manuscript, we have included the following sentence on p. 9, l. 148-149: “No participant referred use of assisted reproductive technology for conception”. 

5. As you mentioned, the crown-rump length (CRL) in the first trimester is a gold-standard for evaluating GA, then why didn’t you measure CRL as a reference method?

Yes, crown-rump length (CRL) up to 84 mm or <14 weeks is regarded as the gold standard for determining gestational age. Unfortunately, we were not able to measure CRL before 14 weeks of gestational age, as 66% of pregnant women reported that the current pregnancy was not planned (Table 2), which commonly incurs in late presentation for starting antenatal visits. In view of these characteristics, second trimester ultrasound examinations were considered a more accurate method for estimating gestational age when compared to LMP, as well as a more feasible procedure to the process of recruiting participants in contrast to CRL. In addressing this point, an explanation has been added on p. 6, l. 98-100, as follows: “Considering characteristics of the study setting on coverage of antenatal care [14], second trimester ultrasound examinations were deemed as more feasible to confirm GA in comparison with CRL”.

6. The regularity of menstruation should be defined.

To address this comment, we have defined the regularity of menstrual cycles on p. 6, l. 93-95, as follows: “Additionally, data on regularity and usual duration of menstrual cycles (regular if usually lasting from 24 to 32 days), as well as type of contraceptive methods in use before conception (if in use, hormonal or not) were collected”.

7. In “data analysis” (page 8, line 139), the types of contraceptive methods in use before conception has been documented but it seems that you only separate the contraception as “hormonal contraceptive use” and “non-hormonal contraceptive use”. I think the emergency contraceptives and short-acting contraceptives are different and the detailed information concerning on contraceptive method should be clarified.

Thank you for this comment. Among types of contraceptive methods before conception, participants (n=505 with LMP and ultrasound data for comparison) reported the use of condoms, oral contraceptive pills, contraceptive injections, hormonal intrauterine devices, and emergency contraceptive pills. Only three participants referred the use of emergency contraceptives. We have categorized contraceptive methods as non-hormonal or hormonal because the use of hormonal contraceptives before conception may indicate poor reliability of self-reported LMP. As recommended, we have detailed the types of contraceptive methods reported by our participants, as well as explained the rationale for the categorization on p. 8-9, l. 142-147: “Confirmation of GA considered the mean ultrasound estimate of GA in relation to information on LMP, regularity of menstrual cycles and type of contraceptive methods in use before conception, which included condoms, oral contraceptive pills, contraceptive injections, hormonal intrauterine devices and emergency contraceptive pills, in isolate or combined use. Contraceptive methods were categorized as non-hormonal or hormonal, as the latter may affect the reliability of self-reported LMP”.

8. A multi-stage algorithm was applied to assess the best estimate of GA during the antenatal period. How did you build this algorithm? Have you done a pilot study?

The algorithm used in the present study was based on a review of guidelines and related literature including the most recent recommendations from the American College of Obstetricians and Gynecologists [i], the Diagnostic Imaging Committee of Canada [ii], the UK National Collaborating Centre for Women’s and Children’s Health [iii] and the Brazilian Ministry of Health [iv]. We also considered procedures adopted in the Intergrowth-21st Project, in the Birth in Brazil Study and in an ultrasound demonstration project in Malawi [v-vii], with adaptations to our study setting and available resources. We have not carried out a pilot study. The basis of the proposed algorithm is indicated on p. 9, l. 156-167, as follows: “A multi-stage algorithm was used to assess the best estimate of GA, according to an evaluation of reliability of LMP and acceptable differences in relation to ultrasound examinations. LMP was considered reliable if reported with certainty, along with regular menstrual cycles and no hormonal contraceptive use before conception [19]. After a review of guidelines and related literature [1, 11, 20-24], acceptable differences of reliable LMP in relation to the ultrasound estimate were defined at ≤7 days for ultrasound examinations conducted up to 22 weeks of pregnancy; for pregnancies with longer durations, differences were acceptable at ≤14 days. Regarding timing of ultrasound data acquisition in the present study, 62.9% of participants were examined up to 20 weeks of pregnancy and 81.9% up to 22 weeks of pregnancy; only 5.0% of examinations were conducted in the third trimester. Such criteria were additionally compared with other definitions for acceptable differences between LMP and ultrasound estimates [1, 21], and our findings were not substantially changed”. 

Cited literature:

[i] American College of Obstetricians and Gynecologists. Methods for estimating the due date. Committee Opinion no. 700. Obstet Gynecol. 2017;129:e150-4.

[ii] Butt K, Lim K; Diagnostic Imaging Committee. Guideline no. 388-Determination of gestational age by ultrasound. J Obstet Gynaecol Can. 2019;41:1497-507.

[iii] National Collaborating Centre for Women’s and Children’s Health (UK). Antenatal care: routine care for the healthy pregnant woman. London: RCOG Press, 2008.

[iv] Brasil. Ministério da Saúde. Secretaria de Atenção à Saúde. Departamento de Atenção Básica. Atenção ao pré-natal de baixo risco. Cadernos de Atenção Básica, n° 32. Brasília, 2012.

[v] Papageorghiou AT, Kemp B, Stones W, Ohuma EO, Kennedy SH, Purwar M, et al. Ultrasound-based gestational-age estimation in late pregnancy. Ultrasound Obstet Gynecol. 2016;48:719-26.

[vi] Pereira APE, Leal MC, Gama SGN, Domingues RMSM, Schilithz AOC, Bastos MH. Determining gestational age based on information from the Birth in Brazil study. Cad Saude Publica. 2014;30:S59-70.

[vii] Wylie BJ, Kalilani-Phiri L, Madanitsa M, Membe G, Nyirenda O, Mawindo P, et al. Gestational age assessment in malaria pregnancy cohorts: a prospective ultrasound demonstration project in Malawi. Malaria J. 2013;12:183.

9. It is necessary to explain how the sample size is calculated.

From February 2015 to January 2016, women up to 20 gestational weeks were recruited while booking an appointment for antenatal care at each of the 13 primary health care units, covering the entire urban area of Cruzeiro do Sul. We estimated to track approximately 850 pregnant women –information on the basis for such estimate has been provided on p. 5, l. 72-79, as follows: “Our study population was composed of pregnant women enrolled in antenatal care in all primary health care units from the urban area (n=13). Women up to 20 weeks of pregnancy, as initially screened by their LMP at the primary health care units, who were living in the city and intended to deliver at the only maternity hospital in Cruzeiro do Sul were considered eligible for this study. We estimated to track approximately 850 pregnant women, considering: (i) the number of deliveries at the local maternity hospital in 2013 (n=1,780), (ii) the percentage of urban population in the municipality (around 60%), and (iii) the coverage for local primary health care services (equivalent to 80%)”. 

10. The present study analyzed the agreement of antenatal estimates of GA confirmed by ultrasound and clinical records at birth. But what is basis of GA from clinical records? Is it evaluated from LMP or checked by the CRL in the first trimester?

Thank you for the comment. In Brazil, LMP is the first fill-in option for calculation of GA according to national guidelines on antenatal care, but ultrasound methods are increasingly available and advised. Therefore, in this study, clinical records at birth as collected from the maternity hospital may in fact combine mixed sources of information. This measure of GA is supplied to health information systems on live births. Such aspects have been included in the discussion of the manuscript, on p. 20, l. 377-384. 

Reviewer #2

1. Overall I believe this is a strong manuscript with multiple strengths -- the use of the algorithm to determine the the validity of LMP measures and the data on image quality. 

Thank you. We sincerely appreciate the reviewer’s evaluation on our manuscript.

2. Poor comma usage throughout the manuscript which made it distracting and difficult to understand at times.

As pointed out by the reviewer, we have revised the comma usage throughout the manuscript. 

3. Several sentences needed to be reworded for clarity [L95-97] and [L339-341].

We have amended these sentences for conciseness, as follows: 

- “Three field physicians with high-quality training in ultrasound examinations (DLL, EV, RBF) were presented to the study equipment and protocols for image acquisition, storage and extraction” (p. 6, l.100-102);

- “Lee at al. [32] systematically reviewed the literature on the diagnostic accuracy and reliability of GA determination of newborns in relation to ultrasound or LMP. The authors indicated poor quality of studies on GA ascertainment and listed priority actions for low- to middle-income regions, including improvement of coverage and development of novel ultrasound approaches [32]” (p.19, l. 355-359). 

4. The description of the algorithm is a bit confusing/misleading. I really enjoyed the analysis but I think the way it is described doesn't make it clear that the algorithm is validating the utility/accuracy of LMP measures.

Thank you for this comment. As suggested, we have edited the description of the algorithm as follows: “A multi-stage algorithm was used to assess the best estimate of GA, according to an evaluation of reliability of LMP and acceptable differences in relation to ultrasound examinations. LMP was considered reliable if reported with certainty, along with regular menstrual cycles and no hormonal contraceptive use before conception [19]. After a review of guidelines and related literature [1, 11, 20-24], acceptable differences of reliable LMP in relation to the ultrasound estimate were defined at ≤7 days for ultrasound examinations conducted up to 22 weeks of pregnancy; for pregnancies with longer durations, differences were acceptable at ≤14 days” (p. 9, l. 156-162).

5. There were 15 participants that only had ultrasound and no LMP. However, it is unclear how they were eligible as eligibility was based on LMP determined gestational age.

All pregnant women were initially screened according to their LMP while booking an appointment for antenatal care at the primary health care units. Upon acceptance to participate in the MINA-Brazil Study, sociodemographic and health interviews gathered information on regularity of menstrual cycles and use of contraceptive methods, which were essential for evaluating reliability of LMP according to our algorithm. Then, ultrasound examinations were scheduled. Overall, the reported LMP of 14 pregnant women overestimated their GA and during the ultrasound examination field physicians could measure a crown-rump length up to 84 mm. As this is considered the gold standard for determining GA, we assumed this information as the best estimate of GA for these participants. For one additional participant, data on menstrual cycles and contraceptives were missing from the interview, but she could attend the ultrasound examination. To clarify this point, the data analysis section depicts the following sentence: “Among 14 participants, a fetal crown-rump length <84 mm was noted at the moment of the first ultrasound examination during follow-up assessments and the corresponding GA was set as the best available estimate in the antenatal period. One additional participant did not provide information on menstrual cycles and contraceptives; therefore, the ultrasound estimate of GA was solely used” (p. 9-10, l. 167-171).

6. The relationship between education and cesarean section was not unexpected. However, the greater difference in LMP and ultrasound in those groups was. I think the potential reasons for this should be discussed.

We agree with the reviewer that the association between education level and occurrence of caesarean sections was expected, as well documented in the literature. In fact, in our analysis there was no comparison between LMP and ultrasound estimates according to education level or type of delivery, because data on LMP were checked for reliability and used in the multistage algorithm in order to assess the best antenatal estimate of GA. Subsequently, we could observe higher mean differences in GA in agreement analysis between antenatal estimates by ultrasound and clinical records at birth for participants with higher education levels and cases of caesarean sections. We have discussed potential reasons for these findings on p. 20-21 (in paragraphs between l. 385 and l. 406), pointing out to the elective use of this health technology among women of higher socioeconomic status in the complex scenario documented in Brazil of elevated caesarean sections rates in conjunction with an increasing proportion of early term births. 

7. I think Figure 3 would be easier to interpret if the two histograms were overlaid.

As suggested, we have reformulated Figure 3 in order to present both histograms side by side. Minor edits have been performed in the text on p. 14 to refrain from referring to Fig 3A and Fig 3B.

8. Was there any proportional bias in the Bland-Altman plots ie a relationship between difference in measures for different gestational ages?

Thank you for this comment, which was quite important to strengthen the presentation and discussion of our findings. We have reviewed the Bland-Altman analysis and investigated proportional bias by regressing differences between antenatal estimates by ultrasound and clinical records at birth on averages of GA from these sources of information, as included on p. 10, l. 181-182: “Linear regression of differences on averages was fitted to investigate proportional bias, detected when the coefficient is significantly different from zero”. After performing this additional step, we amended our results and discussion sections, as follows:

- “The linear regression coefficient of difference on mean was equal to 0.18 (95% CI: 0.11, 0.25), indicating significant proportional bias. Visual inspection of the Bland-Altman plot suggests that wider differences were observed at higher mean values of GA between measurements. This finding indicates that, at the higher end of the distribution of GA, clinical records at birth registered lower values compared with antenatal data, which is consistent to some extend with the small proportion of postterm births according to clinical records and the poor agreement between both sources of information for such classification” (p. 15, l. 267-273);

- “Proportional bias was found, suggesting that wider differences between these sources of information for GA were observed at higher mean values of GA between measurements” (p.16-17, l. 307-309);

- “Distribution of values according to clinical records at birth do not appear sufficiently dispersed to the right, remarkably after 40 weeks, in a different pattern in relation to antenatal estimates by ultrasound, which resulted in significant proportional bias and inconsistent agreement for the classification of postterm births in our study” (p. 20, l. 385-388).

9. I think it would be useful to know how the agreement between LMP and ultrasound estimates differed based on the gestational age at ultrasound.

As indicated in answering comment #6 from the reviewer, data on LMP were checked for reliability and used in the multistage algorithm with ultrasound data, in order to assess the best antenatal estimate of GA. Agreement analysis was then performed between the antenatal estimates of GA by ultrasound and the information on GA as available in clinical records at birth.

---

## [Editor Report · Decision Letter 1]

29 Jun 2020

Agreement between antenatal gestational age by ultrasound and clinical records at birth: a prospective cohort in the Brazilian Amazon

PONE-D-20-08619R1

Dear Dr. Lourenco,

We’re pleased to inform you that your manuscript has been judged scientifically suitable for publication and will be formally accepted for publication once it meets all outstanding technical requirements.

Kind regards,

Professor Kwasi Torpey, MD PhD MPH

Academic Editor

PLOS ONE
---

## [Editor Report · Acceptance letter]

1 Jul 2020

PONE-D-20-08619R1 

Agreement between antenatal gestational age by ultrasound and clinical records at birth: a prospective cohort in the Brazilian Amazon 

Dear Dr. Lourenço:

I'm pleased to inform you that your manuscript has been deemed suitable for publication in PLOS ONE. Congratulations! Your manuscript is now with our production department. 

Kind regards, 

on behalf of

Professor Kwasi Torpey 

Academic Editor

PLOS ONE